# Optimal Rates for Random Order Online Optimization

**Uri Sherman**
Blavatnik School of Computer Science
Tel Aviv University
urisherman@mail.tau.ac.il.

**Tomer Koren**
Blavatnik School of Computer Science
Tel Aviv University, and Google Research
tkoren@tauex.tau.ac.il.

**Yishay Mansour**
Blavatnik School of Computer Science
Tel Aviv University, and Google Research
mansour.yishay@gmail.com.

## Abstract

We study online convex optimization in the random order model, recently proposed by Garber et al. [8], where the loss functions may be chosen by an adversary, but are then presented to the online algorithm in a uniformly random order. Focusing on the scenario where the cumulative loss function is (strongly) convex, yet individual loss functions are smooth but might be non-convex, we give algorithms that achieve the optimal bounds and significantly outperform the results of Garber et al. [8], completely removing the dimension dependence and improving their scaling with respect to the strong convexity parameter. Our analysis relies on novel connections between algorithmic stability and generalization for sampling without-replacement analogous to those studied in the with-replacement i.i.d. setting, as well as on a refined average stability analysis of stochastic gradient descent.

## 1 Introduction

Online convex optimization [25, 12] studies the iterative process of decision making as data arrives in an online fashion. The model posits a game of $T$ rounds, where in each round the learner chooses a decision $w_t$ from a convex set $W \subseteq \mathbb{R}^d$, after which she observes a loss function $f_t : W \to \mathbb{R}$, and incurs loss $f_t(w_t)$. The learner's objective is to minimize her regret, defined as her cumulative loss minus that of the best decision in hindsight $w^* = \arg\min_{w \in W} \sum_{t=1}^{T} f_t(w)$. In the prototypical setting, the individual loss functions are assumed to be convex and adversarially chosen by an opponent—commonly known as nature or the adversary—who has knowledge of the learner's algorithm. While this setup is fundamental enough to accommodate a diverse set of applications (see, e.g., [12]), studying variants of the basic model promotes modeling flexibility, and further broadens the set of problems to which optimization techniques may be applied.

Recently, Garber et al. [8] consider relaxing the convexity assumption by requiring that only *on average* the loss is (strongly) convex—a property the authors refer to as cumulative (strong) convexity—but do not require that the losses are convex individually. It is well known (e.g., [4]) that under these assumptions, if the losses are sampled i.i.d. from some distribution, stochastic gradient descent (SGD) obtains the optimal $O(\log T)$ regret in expectation. However, as it turns out, in the fully adversarial model the cumulative strong convexity assumption is too weak: Garber et al. [8] show that in this case there is a linear regret lower bound. Consequently, they propose the *random order* model, where $T$ losses are chosen adversarially but then revealed to the learner in uniformly random order. Within this model, under the relaxed convexity assumption Garber et al. [8] obtain sub-linear regret for a number of specialized settings, differing in their assumptions on the structure of the individual loss functions.

35th Conference on Neural Information Processing Systems (NeurIPS 2021).

In the most general case (the one we consider in this paper), they prove online gradient descent obtains regret $O((dG^2/\lambda^3)\log T)$ w.h.p. for $G$-Lipschitz $\lambda$-cumulative-strongly convex losses.

It is informative to compare the random order model with the i.i.d. stochastic case, where on every round a new loss is sampled uniformly and *independently* from the set of losses, that is, *with-replacement*. By contrast, the random order model specifies that on every round a new loss is sampled uniformly *without-replacement*, an in particular *not independently*. Concretely, let $\mathcal{L}$ be an arbitrary set of $T$ smooth and Lipschitz continuous loss functions. Set $F(w) \coloneqq \frac{1}{T}\sum_{f \in \mathcal{L}} f(w)$, and assume $F$ is $\lambda$-strongly-convex over $W$. The learner's goal is to minimize her regret on the loss sequence $f_1, \ldots, f_T$ obtained from a uniformly random ordering of $\mathcal{L}$. As noted previously, if the losses $f_t$ were drawn i.i.d., SGD obtains the optimal $O(\log T)$ regret even though the losses are not individually convex. In a nutshell, when losses are i.i.d., the gradients used in the SGD update are conditionally unbiased estimates of the gradient of the average (strongly convex) loss, hence the optimal regret is achieved in expectation. The difficulty in the random order model stems from the fact that random order gradients are not conditionally unbiased; given any set of past losses $f_1, \ldots, f_{t-1}$, the next loss is uniform over the complement $\mathcal{L} \setminus \{f_1, \ldots, f_{t-1}\}$, and thus $\nabla f_t(w_t)$ is biased.

To overcome this complication, Garber et al. [8] work via the uniform convergence route, and build on concentration bounds applied to Hessians of the losses. As a result, they achieve suboptimal bounds, particularly in the general case where a dimension factor is introduced by a discretization argument necessary to ensure convergence over the entire domain. Here we choose a different strategy, and draw connections to notions of algorithmic stability and generalization studied in statistical learning theory. Our approach introduces significant improvements compared to prior work, and achieves regret bounds optimal up to additive factors.

## 1.1 Our results

We present and analyze two algorithms for random order online optimization, the first of which obtains the optimal regret up to additive factors. Let $f_1, \ldots, f_T$ be a random order sequence of $G$-Lipschitz, $\beta$-smooth losses, where $f_t : W \to \mathbb{R}$ for all $t \le T$. Assume the domain $W \subset \mathbb{R}^d$ is convex, and has diameter bounded by $D$. Further, assume the average loss $\frac{1}{T}\sum_{t=1}^{T} f_t$ is $\lambda$-strongly convex. We prove;

**Theorem** (informal). *There exists an algorithm (Algorithm 1) for random order online optimization that obtains regret of $O\left((G^2/\lambda)\log T\right)$ in expectation.*

Although the above result matches the optimal result for the *individually* strongly convex setting (up to additive factors), Algorithm 1 requires memory linear in $T$. This disadvantage motivates another algorithm, which trades off an extra factor of $\kappa = \beta/\lambda$ in the regret for lower memory requirements.

**Theorem** (informal). *There exists an algorithm (Algorithm 2) for random order online optimization that requires memory linear in d, and obtains regret of $O\left((\beta G^2/\lambda^2)\log T\right)$ in expectation.*

Big-$O$ notation in both theorems hides additive factors polynomial in problem parameters $\beta$, $G$ and $D$ (but does not hide multiplicative factors in these parameters). By comparison, Garber et al. [8] obtain a regret bound of $O((dG^2/\lambda^3)\log T)$ w.h.p. for this setting. Both of our algorithms completely remove the dimension factor $d$, and reduce scaling w.r.t. the strong convexity parameter by a factor of $1/\lambda^2$ and $1/\lambda$ respectively. Moreover, their results require the losses to have a Lipschitz Hessian (see 8, Theorem A.3), an assumption we do not make.

In addition, we consider the case that $F$ is convex (but not strongly convex), and apply our above results by means of regularization. As a corollary of the first theorem, we obtain for this setting regret that scales as $\widetilde{O}(\sqrt{T})$, matching up to logarithmic and additive factors the optimal rate for the *individually* convex setting. Similarly, the second theorem implies a $\widetilde{O}(T^{2/3})$ regret algorithm which is also memory efficient for the convex $F$ case. Notably, a similar reduction applied to the results of Garber et al. [8] would yield a regret bound of $\widetilde{O}(dT^{3/4})$. This highlights the significance of our improvement to the dependence on $\lambda$.

Finally, we note it remains an open question to determine whether vanilla without-replacement SGD achieves optimal regret in the random-order model (our results leave a multiplicative gap of order $1/\lambda$ between the upper and lower bounds). For the case of quadratic losses, however, using our techniques combined with a concentration argument it is possible to show that SGD indeed achieves optimal $O(\log T/\lambda)$ regret bound (up to logarithmic terms).

## 1.2 Overview of techniques

Our approach builds on the observation that regret on a random order loss sequence may be expressed as the average generalization error w.r.t a without-replacement training sample. In light of this, we relate random order regret to a suitable notion of algorithmic stability, mimicking in a sense a well known argument previously employed in the context of i.i.d. sampled training sets [3, 21]. At a high level, stability measures the sensitivity of an algorithm to small changes in its training set, and is a classical approach to proving generalization bounds [3, 21]. More often than not, the particular notion used in practice is *uniform* stability, where sensitivity is measured w.r.t. the worst case small change in the training dataset; $\max_{S \subseteq \mathcal{Z}(m)} \|f(\mathcal{A}(S) - f(\mathcal{A}(S')))\|$.

Our key insight is that while we cannot hope for uniform stability as losses are not assumed to be convex individually, we may exploit strong convexity of the population loss to show SGD admits *average* stability; $\mathbb{E}_{S \sim \mathcal{Z}(m)} \|f(\mathcal{A}(S)) - f(\mathcal{A}(S'))\| \leq \epsilon(m)$. In particular, we prove the gradient update is *contractive in expectation*; $\mathbb{E} \|x - \eta \nabla f(x) - (y - \eta \nabla f(y))\| \lesssim \mathbb{E} \|x - y\|$ under the cumulative strong convexity assumption. In turn, this yields a stability result which implies regret that scales with $1/t$ for the early rounds up to $t \approx T/\kappa$, where $\kappa$ is the condition number of the problem. In short, as the game progresses the bias of the random order gradient estimates increases, and the gradient update becomes unstable.

To overcome the loss of stability in later rounds, we devise a simple online sampling mechanism that generates i.i.d. uniform samples from a random order distribution, effectively ensuring unbiased gradient estimates throughout all $T$ rounds, and consequently optimal regret up to additive factors. Finally, our approach allows us to develop an analysis framework that naturally accommodates SGD based algorithms in the random order model, and in a broader sense establish stability of SGD in a new, relatively general setting.

## 1.3 Related work

Random-Order Online Optimization was proposed in the recent work of [8] (where it is referred to as ROOCO), who establish an $O((dG^2/\lambda^3) \log T)$ regret upper bound for $G$-Lipschitz $\lambda$-cumulative-strongly convex losses. In the classical OCO setup [25, 12], under the assumption the losses are $\lambda$-strongly convex individually, it is well known the minimax regret scales as $\Theta((G^2/\lambda) \log T)$, and that the lower bound also applies under the assumption the adversary is i.i.d. stochastic [14, 13]. [1] In addition, it is well known (see e.g., [4]) that the upper bound for the i.i.d. adversary is obtained in expectation by SGD also for non-convex losses, as long as the expected loss is strongly convex. In this work, we show that the same regret upper bound also holds for the random order adversary.

Also relevant to our work is the study of stability and generalization [3, 21] in modern learning theory, and in particular stability properties of SGD. Proving generalization bounds with stability arguments is a well known approach dating back at least to [19, 5, 6]. The specific notion of average stability in the i.i.d. setting we draw upon was defined in [21], though many similar measures have appeared in the literature long before their work (see [3, 21] for an overview). The influential work of [11] gave the first generalization bounds for general forms of SGD with an analysis relying on the notion of uniform stability [3]. Since then, several works have used a similar approach to gain further insight into stability and generalization properties of SGD, e.g., [15, 7, 2].

A related line of work [9, 22, 16, 20, 18] studies SGD without-replacement for solving finite-sum optimization problems, a setting commonly encountered in offline machine learning applications. Here, multiple epochs of SGD are executed over a given training set, with the objective to produce a single output (approximately) minimizing the average loss. However, the majority of the results are obtained under the (vastly simplifying) assumption that the individual loss functions are convex, and therefore do not apply in our setting. In addition, the performance metric of interest is convergence rate, and not regret which is the focus of our paper. In particular, Nagaraj et al. [16] employ the method of exchangeable pairs to relate the average and random order loss to a stability-like property, and obtain optimal (up to polylogarithmic factors) convergence rate for a single epoch, albeit only for individually convex loss functions.

Recently, a number of papers study SGD without-replacement and attempt to relax the convexity assumption [10, 17, 1], but the bounds they obtain are under conditions inapplicable for our setting.

---

[1] The lower bound given in Hazan and Kale [13] uses quadratic (hence, smooth) losses, and is easily extensible to without-replacement sampling, and therefore applies in our setting.

Specifically, they impose a requirement that the number of epochs passes a certain threshold strictly larger than one. Moreover, state-of-the-art bounds achieved by [1] are suboptimal w.r.t ours even had we ignored the epoch requirement.

## 2 Setup: Random-Order Online Optimization

In this section, we review notation and assumptions used throughout the paper, and give the formal definition of the model we consider. We let $Z = \{\zeta_1, \ldots, \zeta_T\}$ denote an arbitrary set of $T$ different datapoints, and denote by $W \subseteq \mathbb{R}^d$ a closed convex set with diameter bounded by $D := \max_{x,y \in W} \|x - y\|$. In addition, we let $\Pi(x) := \Pi_W(x) := \arg\min_{w \in W} \|x - w\|^2$ denote the orthogonal projection onto $W$.

We consider a loss function $f : W \times Z \to \mathbb{R}$, and denote the average (also expected / population) loss by $F(w) := \frac{1}{T} \sum_{z \in Z} f(w; z)$. We make the following assumptions;

**Assumption 1** (Individual Lipschitz Continuity). For all $z \in Z$, $f(\cdot; z)$ is $G$-Lipschitz; namely $\|\nabla f(w; z)\| \leq G$ for all $w \in W$.

**Assumption 2** (Individual smoothness). For all $z \in Z$, $f(\cdot; z)$ is $\beta$-smooth; namely $\|\nabla f(x; z) - \nabla f(y; z)\| \leq \beta \|x - y\|$ for all $x, y \in W$.

**Assumption 3** (Cumulative strong convexity). $F$ is $\lambda$-strongly convex; namely $F(y) \geq F(x) + \nabla F(x)^\top (y - x) + \frac{\lambda}{2} \|y - x\|^2$ for all $x, y \in W$.

In addition, we define the condition number of $F$ by $\kappa := \beta / \lambda$.

We will be primarily interested in the random sequence of losses $f(\cdot; z_t)$ obtained from uniformly random orderings of $Z$. Let $z_1, \ldots, z_m \sim \mathcal{Z}(m)$ denote a random sequence of $m$ datapoints, where $z_t = \zeta_{\sigma_t} \in Z$ and $\sigma : [T] \to [T]$ is a uniformly random permutation. Equivalently, $\mathcal{Z}(m)$ may be also considered as the distribution of $m$ datapoints sampled sequentially without-replacement from $Z$. Omitting the number of samples parameter and writing $z \sim \mathcal{Z}$ denotes a uniformly random sample of a single datapoint from $Z$. In addition, if $S = (z_1, \ldots, z_m)$ is a sequence of datapoints, $z \sim \mathcal{Z} \setminus S$ denotes a uniformly random sample of a single datapoint from $Z \setminus S$. In sake of conciseness, we write $S, \tilde{z} \sim \mathcal{Z}(m, 1)$ to denote a sample of a sequence $S \sim \mathcal{Z}(m)$, followed by a sample from the complement $\tilde{z} \sim \mathcal{Z} \setminus S$. We write $\mathcal{F}_m = \sigma(z_1, \ldots, z_m)$ to denote the filtration given by the sequence of $\sigma$-algebras generated by the random variables $(z_1, \ldots, z_m)$. This, of course, is equivalent to sampling $z_1, \ldots z_{m+1} \sim \mathcal{Z}(m + 1)$, and then setting $S = (z_1, \ldots, z_m)$ and $\tilde{z} = z_{m+1}$.

Given a random order sequence $z_1, \ldots, z_T \sim \mathcal{Z}(T)$, we consider the problem of minimizing the expected regret with an online algorithm. We denote the minimizer of the population loss $F$ by $w^* := \arg\min_{w \in W} F(w)$, and let $w_t$ denote the iterates produced by an online algorithm $\mathcal{A}$. The expected regret of $\mathcal{A}$ on $\mathcal{Z}(T)$ is defined as;

$$\mathcal{R}_T := \mathbb{E}\Big[ \sum_{t=1}^{T} f(w_t; z_t) - f(w^*; z_t) \Big],$$

where the expectation is over the random order sequence and any randomness potentially introduced by $\mathcal{A}$. Finally, when a sequence of realized datapoints $z_1, \ldots, z_m$ is clear from context, we let $F_m(w) := \frac{1}{m} \sum_{t=1}^{m} f(w; z_t)$ denote their empirical average loss.

## 3 Stability and Generalization Without Replacement

In this section, we discuss notions of stability and generalization when sampling without-replacement, and give basic results relating to stability of SGD in the setting under consideration. We work with ordered training sets, and write $S = (z_1, \ldots, z_m)$ to make the ordering explicit in our notation. When such a training set is in context along with another datapoint $\tilde{z}_i$, we define $S^{(i)} := (z_1, \ldots, z_{i-1}, \tilde{z}_i, z_{i+1}, \ldots, z_m)$ to be the new training set formed by taking $S$ and swapping the $i$'th datapoint $z_i$ with $\tilde{z}_i$. Finally, we say $\mathcal{A}$ is a learning algorithm if it maps training sets of any length to a decision; $\mathcal{A} : Z^* \to W$.

### 3.1 Recap: Stability and generalization in the i.i.d. setting

In this section we recall the relevant definitions previously studied in the i.i.d. setting. Here, we assume the training set $S = (z_1, \ldots, z_m) \sim \mathcal{D}^m$ is an i.i.d. sample of $m$ datapoints from some predefined distribution $\mathcal{D}$ over elements of $Z$.

**Definition** (on-average generalization; [21, 3]). We say a learning algorithm $\mathcal{A}$ on-average-generalizes with rate $\epsilon_{\text{gen}}(m)$ if for all $m$;

$$|\mathbb{E}_{S \sim \mathcal{D}^m}[F_m(\mathcal{A}(S)) - F(\mathcal{A}(S))]| \leq \epsilon_{\text{gen}}(m).$$

The definition of stability that follows relates a small change in the training set $S \to S^{(i)}$ to the change in the learning algorithm's performance. Here, as one would expect, the swapped datapoint $\tilde{z}_i \sim \mathcal{D}$ is sampled independently from the original training set sample $S$.

**Definition** (average-RO stability; [21, 3]). We say a learning algorithm $\mathcal{A}$ is average-replace-one stable with rate $\epsilon_{\text{stab}}(m)$ if for all $m$;

$$\left| \frac{1}{m} \sum_{i=1}^{m} \mathbb{E}_{S \sim \mathcal{D}^m, \tilde{z}_i \sim \mathcal{D}}[f(\mathcal{A}(S); \tilde{z}_i) - f(\mathcal{A}(S^{(i)}); \tilde{z}_i)] \right| \leq \epsilon_{\text{stab}}(m).$$

With the above definitions, it is well known stability and generalization are in fact equivalent (e.g., [21]).

### 3.2 Stability and generalization without replacement

In this section we discuss the analogous notions suitable for sampling without-replacement. We adopt the term out-of-sample (oos) to distinguish the without-replacement setting, and say $\mathcal{A}$ on-average-generalize-oos with rate $\epsilon_{\text{gen}}(m)$ if

$$\left| \mathbb{E}_{S, \tilde{z} \sim \mathcal{Z}(m,1)} \left[ f(\mathcal{A}(S); \tilde{z}) - F_m(\mathcal{A}(S)) \right] \right| \leq \epsilon_{\text{gen}}(m). \tag{1}$$

Note that here, generalization is measured w.r.t. a datapoint drawn out-of-sample, and in particular *not independently* of $S$. The situation is similar for the notion of stability; while in the i.i.d. case the "non-coupled" index $i$ in $S^{(i)}$ hosts a different datapoint sampled independently, here this datapoint is sampled from the complement $Z \setminus S$. The analogous definition for stability without-replacement says a learning algorithm $\mathcal{A}$ is average-replace-one-oos stable with rate $\epsilon_{\text{stab}}$ if

$$\left| \frac{1}{m} \sum_{i=1}^{m} \mathbb{E}_{S, \tilde{z}_i \sim \mathcal{Z}(m,1)}[f(\mathcal{A}(S); \tilde{z}_i) - f(\mathcal{A}(S^{(i)}); \tilde{z}_i)] \right| \leq \epsilon_{\text{stab}}(m). \tag{2}$$

However, it will be more convenient in our case to work with a slightly different definition, which relates to the distance between *outputs* of the learning algorithm, rather than to the change in out-of-sample loss. We consider w.l.o.g. *randomized* learning algorithms $A : Z^* \times \mathcal{X} \to W$, where $\xi \in \mathcal{X}$ denotes the internal random seed used by $\mathcal{A}$. For convenience, we slightly overload notation and let $\xi \sim \mathcal{X}$ denote the distribution over $\mathcal{A}$'s random seeds.

**Definition 1** (on-average-oos stability). A learning algorithm $\mathcal{A}$ is on-average-oos stable with rate $\epsilon_{\text{stab}}(m)$ on random order distribution $\mathcal{Z}$ if

$$\max_{i \leq m} \mathbb{E}_{S, \tilde{z}_i \sim \mathcal{Z}(m,1), \xi \sim \mathcal{X}} \left[ \|\mathcal{A}(S; \xi) - \mathcal{A}(S^{(i)}; \xi)\| \right] \leq \epsilon_{\text{stab}}(m).$$

We wish to draw the reader's attention to two important aspects of the above definition. First, note that the measure is w.r.t. *random* training sets, which significantly differs from uniform stability where worst case training sets are considered. The maximum in the definition relates to the *index* of the swapped sample, and not to the training sets $S, S^{(i)}$. Second, the same random seed $\xi$ is fed to $\mathcal{A}$ on both training sets, that is, we measure the expected distance between outputs subject to a maximal coupling of the algorithm's randomness.

When we discuss an *online* algorithm $\mathcal{A}$ and a random order sequence $z_1, \ldots, z_T \sim \mathcal{Z}(T)$ is in context, we denote by $z_{1:m} = (z_1, \ldots, z_m)$ the prefix of length $m$. In addition, if $w_{m+1} = \mathcal{A}(z_{1:m}; \xi)$, we denote the coupled iterate by

$$w_{m+1}^{(i)} := \mathcal{A}(z_{1:m}^{(i)}; \xi) = \mathcal{A}(z_1, \ldots, z_{i-1}, \tilde{z}_i, z_{i+1}, \ldots, z_m; \xi), \quad \text{where } \tilde{z}_i = z_{m+1}. \tag{3}$$

With this notation, if $\mathcal{A}$ satisfies Definition 1 with rate $\epsilon(m)$, we have $\mathbb{E} \|w_{m+1} - w_{m+1}^{(i)}\| \leq \epsilon(m)$ for all $i \leq m$. Note that an online learning algorithm is nothing more than a learning algorithm that respects the order of the samples it is given as input. To conclude this section, we relate the population and out-of-sample performance gap to stability of the learning algorithm, as provided by the below lemma.

**Lemma 1.** *Assume $\mathcal{A}$ is on-avg-oos stable with rate $\epsilon(m)$, and let $\ell : W \times Z \to \mathbb{R}$ be any L-Lipschitz loss function. Then;*

$$\left| \mathbb{E}_{S, \tilde{z} \sim \mathfrak{Z}(m,1), z \sim \mathfrak{Z}} \left[ \ell(\mathcal{A}(S); \tilde{z}) - \ell(\mathcal{A}(S); z) \right] \right| \leq \frac{Lm}{T} \epsilon(m).$$

*If $\mathcal{A}$ is an online algorithm producing iterates $w_t$ and $z_1, \ldots, z_t \sim \mathfrak{Z}(t)$ is a random order sample, this immediately implies*

$$\mathbb{E}[f(w_t; z_t) - F(w_t)] \leq \frac{G(t-1)}{T} \epsilon(t-1).$$

We provide the proof in the full version of the paper [23].

### 3.3 Average stability of SGD

In this section, we develop the basic tools employed to establish that under appropriate conditions, SGD is algorithmically stable in the sense of Definition 1. Crucially, by considering *average* stability we are able to leverage strong convexity of the expected function $F$ and prove the desired result. For $w \in W$ and $\psi : W \to \mathbb{R}$, we denote by $\mathcal{G}(w; \psi, \eta) = \Pi(w - \eta \nabla \psi(w))$ a projected gradient descent step from $w$. Our key lemma stated below and proved in the full version of the paper [23], says a gradient step on a random function $\psi$ is contractive in expectation when $\psi$ is strongly convex in expectation, and the step-size is sufficiently small.

**Lemma 2.** *Consider an arbitrary discrete distribution $\mathcal{P}$ of G-Lipschitz and $\beta$-smooth functions $\psi : W \to \mathbb{R}$ such that $\Psi(w) := \mathbb{E}\psi(w)$ is $\mu$-strongly-convex. Then for any $x, y \in W$, a gradient descent step with step-size $\eta \leq \mu/\beta^2$ satisfies;*

$$\mathbb{E}_{\psi \sim \mathcal{P}} \|\mathcal{G}(x; \psi, \eta) - \mathcal{G}(y; \psi, \eta)\| \leq \left(1 - \frac{\eta\mu}{2}\right) \|x - y\|.$$

Notice that the above result dictates for the step-size to be lesser than $\mu/\beta^2$, which is roughly a factor of $1/\kappa$ smaller than needed to ensure (deterministic) contractivity under the assumption of individually strongly convex losses (see [11]). Lemma 2 serves as a building block to prove stability of SGD subject to relatively generic conditions, which we do next. Loosely speaking, if two training sequences do not differ too much, and the conditional expected loss is strongly convex, the expected distance between SGD iterates shrinks proportionally to the number of iterations executed.

We denote by $\mathrm{GD}(S; w_1, m, \{\eta_t\})$ the iterate $w_{m+1} \in W$ produced by executing $m$ projected gradient descent steps on a sequence of datapoints $S = (z_1, \ldots, z_\tau), \tau \geq m$, starting at the initial point $w_1 \in W$, with step-sizes $\{\eta_t\}$. When any one of $w_1, m$ or $\{\eta_t\}$ are clear from context, they may be omitted in sake of conciseness. Our next lemma quantifies how small perturbations in random training sets translate to the expected change in outputs of SGD. Informally, it says that under the appropriate conditions, if we swap just a single training index $i \leq m$, the sequence of gradient steps arrives at the same output up to distance $O(1/m)$.

**Lemma 3.** *Let $i \leq m \in \mathbb{N}$, and $S = (z_1, \ldots, z_m), S' = (z'_1, \ldots, z'_m)$ be two random datapoint sequences. Further, assume that for $0 < \mu$ and $0 \leq \delta \leq \mu/2\beta$, it holds that*

(i) $\Pr(z_t \neq z'_t \mid \mathcal{F}_{t-1}) \leq \delta$ *for all $t \neq i$;*

(ii) $\mathbb{E}\left[f(w; z_t) \mid \mathcal{F}_{t-1}\right]$ *is $\mu$-strongly convex as a function of $\mathcal{F}_{t-1}$-measurable $w \in W$,*

*where $\mathcal{F}_{t-1} := \sigma(z_1, z'_1, \ldots, z_{t-1}, z'_{t-1})$. Then, for step-size schedule $\eta_t = \min\{\tilde{\mu}/\beta^2, 2/\tilde{\mu}t\}$ with $\tilde{\mu} := \mu - \delta\beta$, and any $w_1 \in W$, we have;*

$$\mathbb{E}\|\mathrm{GD}(S; w_1, m) - \mathrm{GD}(S'; w_1, m)\| \leq \frac{4G}{\tilde{\mu}m}(1 + 4\delta m).$$

The proof of the above lemma is given by following the recursive relation specified by Lemma 2 and the conditions imposed on the random sequences $S, S'$. The details are rather technical and are thus deferred to the full version of the paper [23]. To conclude this section, we state and prove a simple corollary of Lemma 3, establishing stability of SGD when the number of steps taken is sufficiently small.

**Corollary 1.** *Let $w_1 \in W$, and set $\eta_t = \min\left\{\lambda/2\beta^2, 4/\lambda t\right\}$. Then the SGD update defined by $w_{t+1} = \Pi(w_t - \eta_t \nabla f(w_t; z_t))$ is on-avg-oos stable with rate*

$$\epsilon_{stab}(m) \leq \frac{8G}{\lambda m},$$

*for all $m \leq T/2\kappa$.*

*Proof.* Fix $i \leq m$, let $\mathcal{A}$ denote the SGD algorithm, and consider a random order sequence $z_1, \ldots, z_T \sim \mathcal{Z}(T)$. We have $w_{m+1} = \mathcal{A}(z_{1:m})$, and $w_{m+1}^{(i)} = \mathcal{A}(z_{1:m}^{(i)})$ as defined in Eq. (3) (note that here though, $\mathcal{A}$ has no internal randomness). Next, we verify conditions for Lemma 3 are satisfied with the two sequences $S := z_{1:m}$ and $S' := (z_1' \ldots, z_m') := z_{1:m}^{(i)}$.

Let $\mathcal{F}_{t-1} := \sigma(z_1, z_1', \ldots, z_{t-1}, z_{t-1}')$, and by definition of $S$ and $S'$, we have that $z_t = z_t'$ for all $t \neq i$, hence clearly $\Pr(z_t \neq z_t' \mid \mathcal{F}_{t-1}) = 0$ for all $t \neq i$. For the second condition, let $\mathsf{F}_{t-1}$ denote the set of all datapoints observed prior to round $t$; $\mathsf{F}_{t-1} := \{z \in Z \mid z \in (z_1, z_1', \ldots, z_{t-1}, z_{t-1}')\}$. This means $\mathsf{F}_{t-1}$ contains exactly $\{z_1, \ldots, z_{t-1}\}$, and perhaps $\tilde{z}_i$ depending on whether $t > i$, hence $k := |\mathsf{F}_{t-1}| \in \{t, t-1\}$. Now, given $\mathcal{F}_{t-1}$, we have that $z_t$ is uniform over $\mathcal{Z} \setminus \mathsf{F}_{t-1}$, therefore

$$\mathbb{E}[f(w; z_t) \mid \mathcal{F}_{t-1}] = \frac{1}{T-k} \sum_{z \in Z \setminus \mathsf{F}_{t-1}} f(w; z) = \frac{T}{T-k}\left(F(w) - \frac{1}{T}\sum_{z \in \mathsf{F}_{t-1}} f(w; z)\right).$$

By our smoothness assumption, $\frac{1}{T}\sum_{z \in \mathsf{F}_{t-1}} f(w; z)$ is $(k\beta/T)$-smooth, and in addition $k\beta/T \leq m\beta/T \leq (\lambda/2\beta)\beta = \lambda/2$. Therefore, by $\lambda$-strong convexity of $F$ we get that the last term in the above derivation is at least $\lambda/2$-strongly-convex (this follows from a standard argument, see Lemma 10). Therefore, by Lemma 3 with $\mu := \lambda/2, \delta := 0$ it now follows that

$$\mathbb{E}\|w_{m+1} - w_{m+1}^{(i)}\| = \mathbb{E}\|\mathrm{GD}(S) - \mathrm{GD}(S')\| \leq \frac{8G}{\lambda m},$$

which completes the proof. $\qquad\square$

## 4 SGD for Random Order Online Optimization

In this section, we present two algorithms for random order online optimization. Lemma 1 motivates us to derive SGD based algorithms that obtain low regret w.r.t. the population loss $F$, *and* are algorithmically stable in the sense of Definition 1. Given a random order sequence $z_1, \ldots, z_T \sim \mathcal{Z}(T)$, the SGD update with gradient estimates $\{\hat{g}_t\}$ and step sizes $\{\eta_t\}$ is given by

$$w_{t+1} \leftarrow \Pi(w_t - \eta_t \hat{g}_t).$$

It is not hard to show that the regret w.r.t. $F$ of SGD is directly related to the error terms introduced by using $\hat{g}_t$ in place of the true population loss gradients $\nabla F(w_t)$. This fact is made formal in the lemma below, which serves as a starting point for the analysis of both algorithms we present. The proof follows from standard arguments, and is deferred to the full version of the paper [23].

**Lemma 4.** *Consider $\tau$ iterations of SGD with gradient estimates $\{\hat{g}_t\}$ and step-size schedule $\eta_t = \min\left\{\tilde{\mu}/\beta^2, 2/(\tilde{\mu}t)\right\}$. We have that the following bound holds with probability one;*

$$\sum_{t=1}^{\tau} F(w_t) - F(w^*) \leq \frac{\beta^2 D^2}{2\tilde{\mu}} + \frac{G^2}{\tilde{\mu}}(1 + \log \tau) + \sum_{t=1}^{\tau}(\nabla F(w_t) - \hat{g}_t)^\mathsf{T}(w_t - w^*). \tag{4}$$

### 4.1 Reservoir SGD

In light of Corollary 1, it is evident that a different strategy is necessary to achieve stability in late rounds of the online game. As a solution, Algorithm 1 presented here employs a sampling procedure reminiscent of reservoir sampling [24], which results in gradient estimates $\hat{g}_t$ that are *conditionally unbiased* estimates of $\nabla F(w_t)$ throughout all $T$ rounds. This ensures the conditionally expected function on every round is strongly convex, thereby implying stability is maintained for the duration of the game. As another implication, the error terms on the RHS of Eq. (4) vanish, which essentially reduces the optimization problem (i.e., regret w.r.t. the population loss $F$) to the i.i.d. setting. Consequently, a regret bound will follow from a batch-to-online conversion supported by Lemma 1 and Lemma 3.

Our first lemma given below, shows that the intermediate sequence of datapoints $z_t'$ generated in line 4 are i.i.d. uniformly distributed, which immediately implies that $\mathbb{E}\hat{g}_t = \nabla F(w_t)$.

---

**Algorithm 1** ReservoirSGD

---
1: **input:** step-sizes $\eta_1, \ldots, \eta_T \in \mathbb{R}_+, w_1 \in W$
2: **for** $t = 1$ to $T$ **do**
3:     Play $w_t$, Observe $z_t$
4:     Set $\hat{g}_t := \nabla f(w_t; z_t')$, where $z_t' = \begin{cases} z_t & \text{w.p. } 1 - \frac{t-1}{T} \\ \text{Unif}(z_1, \ldots, z_{t-1}) & \text{w.p. } \frac{t-1}{T} \end{cases}$
5:     $w_{t+1} \leftarrow \Pi(w_t - \eta_t \hat{g}_t)$
6: **end for**

---

**Lemma 5.** *The $\{z_t'\}$ intermediate sequence produced by Algorithm 1 in line 4 is uniform over Z and i.i.d.;*

$$\forall z \in Z; \quad \Pr(z_t' = z) = \Pr(z_t' = z \mid z_{<t}) = \frac{1}{T}.$$

*Proof.* Fix the first $t - 1$ sampled datapoints $z_1, \ldots, z_{t-1} \sim \mathcal{Z}(t-1)$. Then for all $z \in \{z_1, \ldots, z_{t-1}\}$,

$$\Pr(z_t' = z \mid z_{1:t-1}) = \frac{t-1}{T} \cdot \frac{1}{t-1} = \frac{1}{T}.$$

In addition, since $z_1, \ldots, z_t \sim \mathcal{Z}(t)$, it follows that $z_t$ is uniform over $Z \setminus z_{1:t-1}$. (Note that as we are conditioning on $z_{1:t-1}$, we have that $Z \setminus z_{1:t-1}$ is deterministic.) Hence, for all $z \in Z \setminus z_{1:t-1}$ we have

$$\Pr(z_t' = z \mid z_{1:t-1}) = \frac{T-t+1}{T} \cdot \frac{1}{T-t+1} = \frac{1}{T}.$$

The above implies that $\Pr(z_t' = z \mid z_{1:t-1}) = 1/T$ for all $z \in Z$. Finally, the by the law of total probability;

$$\Pr(z_t' = z) = \mathbb{E}_{z_1, \ldots z_{t-1} \sim \mathcal{Z}(t-1)} \left[ \Pr(z_t' = z \mid z_{1:t-1}) \right] = \frac{1}{T},$$

as desired. $\qquad\square$

Next, we argue Algorithm 1 maintains average stability with the desired rate throughout all $T$ rounds.

**Lemma 6.** *Assume $T \geq 2\beta/\lambda$, and set $\tilde{\mu} := \lambda - \beta/T$. Then Algorithm 1 with step-size schedule $\eta_t = \min\{\tilde{\mu}/\beta^2, 2/(\tilde{\mu}t)\}$ is on-avg-oos stable (Definition 1) with rate*

$$\epsilon_{stab}(m) \leq \frac{40G}{\lambda m}.$$

*Proof.* Let $i \leq m \leq T$, and recall the definition of the coupled iterate in Eq. (3). We have that

$$\mathbb{E} \|w_{m+1} - w_{m+1}^{(i)}\| = \mathbb{E} \|\text{GD}(S') - \text{GD}(S'')\|,$$

where $S' = (z_1', \ldots, z_m')$ and $S'' = (z_1'', \ldots, z_m'')$ denote the intermediate sequences (line 4) produced when running Algorithm 1 on $z_{1:m}$ and $z_{1:m}^{(i)}$ respectively. Now, consider the indexes of datapoints selected by the sampling mechanism on line 4 which we denote by $j_t$, meaning $j_t = l$ when $z_t' = z_l$. Since the same random seed is used for each coupled iterate, we have that the indexes $j_t$ from both execution paths are the same. Next, we will show $S', S''$ satisfy the conditions of Lemma 3.

Indeed, denote by $\mathcal{F}_t$ the filtration encapsulating all randomness (random order and algorithm) up to and including round $t$. Then by Lemma 5 and since $z_{1:m}$ and $z_{1:m}^{(i)}$ differ only at index $i$, we have for any $t > i$;

$$\Pr(z_t' \neq z_t'' \mid \mathcal{F}_{t-1}) = \Pr(j_t = i \mid \mathcal{F}_{t-1}) = \frac{1}{T}.$$

In addition, it trivially follows that $\Pr(z_t' \neq z_t'' \mid \mathcal{F}_{t-1}) = 0$ for all $t < i$. Owed to our assumption on $T$, we have $1/T \leq \lambda/2\beta$, and so the first condition is satisfied. For the second condition, note that again by Lemma 5, $\mathbb{E}[f(w; z_t') \mid \mathcal{F}_{t-1}] = F(w)$ for any $\mathcal{F}_{t-1}$-measurable $w \in W$, which immediately implies $\lambda$-strong convexity of the conditionally expected function.

By the above, we obtain that $S', S''$ follow a distribution satisfying conditions required by Lemma 3 with $\mu := \lambda, \delta := 1/T$, therefore,

$$\mathbb{E} \|\text{GD}(S'; m) - \text{GD}(S''; m)\| \leq \frac{4G}{(\lambda - (1/T)\beta)m} \left(1 + 4\frac{m}{T}\right) \leq \frac{40G}{\lambda m},$$

and we are done. $\qquad\square$

A regret bound for Algorithm 1 readily follows, as we have essentially established both convergence rate and stability of the algorithm. Below, we state and prove our main result concluding this section.

**Theorem 1.** *Running Algorithm 1 with step-size schedule* $\eta_t = \min\left\{\tilde{\mu}/\beta^2, 2/(\tilde{\mu}t)\right\}$ *where* $\tilde{\mu} := \lambda - \beta/T$, *it is guaranteed that;*

$$\mathbb{E}\Big[\sum_{t=1}^{T} f(w_t; z_t) - f(w^*; z_t)\Big] \leq \frac{2G^2}{\lambda}(1 + \log T) + \frac{40G^2 + \beta^2 D^2 + 2\beta GD}{\lambda}.$$

*Proof.* By Lemma 5, we have that for all $t \leq T$,

$$\mathbb{E}[\nabla f(w_t; z_t')]^\top(w_t - w^*)] = \mathbb{E}[\mathbb{E}_t[\nabla f(w_t; z_t')])^\top(w_t - w^*)] = \mathbb{E}[\nabla F(w_t)^\top(w_t - w^*)],$$

where $\mathbb{E}_t[\cdot] = \mathbb{E}[\cdot \mid z_1, z_1', \ldots, z_{t-1}, z_{t-1}']$ denotes the conditional expectation w.r.t. all rounds up to and not including $t$. This implies the gradient error terms on the RHS of Eq. (4) vanish. In addition, note we may assume $T \geq 2\beta/\lambda$, for otherwise it trivially follows that $\sum_{t=1}^{T} f(w_t; z_t) - f(w^*; z_t) \leq GDT \leq \frac{2\beta GD}{\lambda}$. Hence $\delta := 1/T \leq \lambda/2\beta$, and by Lemma 4 we obtain the following bound on the regret w.r.t. the population loss $F$;

$$\mathbb{E}\Big[\sum_{t=1}^{T} F(w_t) - F(w^*)\Big] \leq \frac{\beta^2 D^2}{\lambda} + \frac{2G^2}{\lambda}(1 + \log T).$$

To establish online performance, observe that by Lemma 1 and Lemma 6 we have

$$\mathbb{E}[f(w_t; z_t) - F(w_t)] \leq \frac{G(t-1)}{T}\epsilon_{\text{stab}}(t-1) = \frac{G(t-1)}{T}\frac{40G}{\lambda(t-1)} = \frac{40G^2}{\lambda T}.$$

Therefore,

$$\mathbb{E}\Big[\sum_{t=1}^{T} f(w_t; z_t) - f(w^*; z_t)\Big] = \sum_{t=1}^{T} \mathbb{E}[f(w_t; z_t) - F(w_t)] + \mathbb{E}\Big[\sum_{t=1}^{T} F(w_t) - F(w^*)\Big]$$

$$\leq \frac{40G^2}{\lambda} + \frac{\beta^2 D^2}{\lambda} + \frac{2G^2}{\lambda}(1 + \log T),$$

and the result follows. $\qquad\square$

## 4.2 SGD without replacement

While Algorithm 1 obtains regret which is optimal up to additive factors, it is memory intensive due to the sampling procedure requiring the history of the entire loss sequence. This motivates Algorithm 2 which uses the random order gradients as they arrive, and sacrifices a factor of $\kappa$ in the regret bound for lower memory requirements.

---

**Algorithm 2** SGD-without-replacement

1: **input:** $\tau, T \in \mathbb{N}$, step-sizes $\eta_1, \ldots, \eta_\tau \in \mathbb{R}_+$, $w_1 \in W$
2: **for** $t = 1$ to $\tau$ **do**
3:     Play $w_t$, Observe $z_t$
4:     $w_{t+1} \leftarrow \Pi(w_t - \eta_t \hat{g}_t)$, where $\hat{g}_t := \nabla f(w_t; z_t)$
5: **end for**
6: **for** $t = \tau + 1, \ldots, T$: play $w_t \equiv \bar{w} := \frac{1}{\tau}\sum_{i=1}^{\tau} w_i$.

---

With Corollary 1, we already know Algorithm 2 is stable with rate $O(1/m)$ for $\tau \leq T/2\kappa$, at least for all iterates excluding $\bar{w}$. However, it is not clear a priori whether these iterates also obtain good convergence rate—to prove this we must control the gradient error terms on the RHS of Eq. (4). Evidently, with random order gradient estimates the behavior of these error terms is also related to stability in the same sense that the actual losses are. By Lemma 1 with $\ell(w; z) := \nabla f(w; z_t)^\top(w - w^*)$, it only remains to derive the appropriate Lipschitz constant, which is done in our next lemma. Notably, this means that the two sources of error, gradient estimates and the batch-to-online gap, both hinge on the very same stability property of the algorithm.

**Lemma 7.** *Running Algorithm 2 with a step-size schedule $\eta_t = \min\left\{\lambda/2\beta^2, 4/\lambda t\right\}$ and $\tau = T/2\kappa$, we have that the gradient error terms on the RHS of Eq. (4) are bounded, for all $t \leq \tau$, as*

$$\mathbb{E}[(\nabla F(w_t) - \hat{g}_t)^\top(w_t - w^*)] \leq \frac{8G(G + \beta D)}{\lambda T}.$$

The proof of Lemma 7 is given by arguments following the above discussion, and is deferred to the full version of the paper [23]. Next, we state and prove our main theorem for this section providing the regret guarantees for Algorithm 2.

**Theorem 2.** *Running Algorithm 2 with a step-size schedule $\eta_t = \min\left\{\lambda/2\beta^2, 4/\lambda t\right\}$ and $\tau = T/2\kappa$, it holds that*

$$\mathbb{E}\left[\sum_{t=1}^{T} f(w_t; z_t) - f(w^*; z_t)\right] = O\left(\frac{\beta G^2}{\lambda^2}\log T + \frac{\beta DG}{\lambda} + \frac{\beta^3 D^2}{\lambda^2}\right).$$

The proof of Theorem 2 provided in the full version of the paper [23], hinges on the ingredients discussed thus far, along with the fact that averaging preserves stability.

### 4.3 The convex case

Given Algorithms 1 and 2, we can derive as an immediate corollary regret bounds for the case where the loss function $f$ is only assumed to be convex in expectation, but not strongly convex. This is achieved by adding L2 regularization; pick $w_0 \in W$ arbitrarily, and consider $f^\alpha(w; z) := f(w; z) + \frac{\alpha}{2}\|w - w_0\|^2$. By transforming the gradient estimators $\hat{g}_t \leftarrow \hat{g}_t + \alpha(w_t - w_0)$ in both algorithms, we optimize for the regularized random order loss sequence $f^\alpha(\cdot; z_1), \ldots f^\alpha(\cdot; z_T)$, and obtain regret bounds for suitable choices of $\alpha$.

**Corollary 2.** *Assume the loss function $f : W \times Z \to \mathbb{R}$ is $G$-Lipschitz and $\beta$-smooth for all $z \in Z$ individually (Assumption 1 and Assumption 2), and that $\frac{1}{T}\sum_{z \in Z} f(\cdot; z)$ is convex over $W$. Then, we have the following guarantees for running Algorithms 1 and 2 on the regularized loss sequence:*

(i) *for Algorithm 1 and $\alpha = 1/\sqrt{T}$;* $\mathbb{E}\left[\sum_{t=1}^{T} f(w_t; z_t) - f(w^*; z_t)\right] = \widetilde{O}(\sqrt{T})$,

(ii) *for Algorithm 2 and $\alpha = 1/T^{1/3}$;* $\mathbb{E}\left[\sum_{t=1}^{T} f(w_t; z_t) - f(w^*; z_t)\right] = \widetilde{O}(T^{2/3})$,

*where big-$\widetilde{O}$ hides polynomial dependence on the problem parameters $\beta, G, D$, and logarithmic dependence on $T$.*

*Proof.* To ease notational clutter denote $f_t(w) := f(w; z_t)$ and $f_t^\alpha(w) := f^\alpha(w; z_t) = f_t(w) + \frac{\alpha}{2}\|w - w_0\|^2$. Let $F^\alpha(w) := \frac{1}{T}\sum_{z \in Z} f^\alpha(w; z)$, then $F^\alpha(w) = F(w) + \frac{\alpha}{2}\|w - w_0\|^2$, which implies that $F^\alpha$ is $\alpha$-strongly convex. Therefore,

$$\mathbb{E}\left[\sum_{t=1}^{T} f_t^\alpha(w_t) - f_t^\alpha(w^*)\right] \leq \mathcal{R}_T(\alpha),$$

where $\mathcal{R}_T(\alpha)$ denotes the regret w.r.t. the regularized loss sequence $f_t^\alpha$. In addition;

$$\sum_{t=1}^{T} f_t(w_t) - f_t(w^*) = \sum_{t=1}^{T} f_t(w_t) - f_t^\alpha(w_t) + \sum_{t=1}^{T} f_t^\alpha(w_t) - f_t^\alpha(w^*) + \sum_{t=1}^{T} f_t^\alpha(w^*) - f_t(w^*)$$

$$= \frac{\alpha}{2}\sum_{t=1}^{T}\|w_t - w_0\|^2 + \sum_{t=1}^{T} f_t^\alpha(w_t) - f_t^\alpha(w^*) + \frac{\alpha}{2}\sum_{t=1}^{T}\|w^* - w_0\|^2$$

$$\leq \sum_{t=1}^{T} f_t^\alpha(w_t) - f_t^\alpha(w^*) + \alpha D^2 T$$

$$\leq \mathcal{R}_T(\alpha) + \alpha D^2 T.$$

The result follows after substituting for the values of $\alpha$ specified in the statement of the theorem, and the regret bounds of Theorems 1 and 2. $\qquad\square$

We conclude by noting the above result underscores the importance of obtaining bounds with good dependence on the strong convexity parameter. In particular, only the optimal $1/\lambda$ dependence allows for regularization that guarantees optimal (up to logarithmic factors) performance w.r.t. $T$ in the non-strongly convex setting.

**Acknowledgements and funding disclosure**

This work was supported by the European Research Council (ERC) under the European Union's Horizon 2020 research and innovation program (grant agreement No. 882396), by the Israel Science Foundation (grants number 993/17 and 2549/19), by the Len Blavatnik and the Blavatnik Family foundation, and by the Yandex Initiative in Machine Learning at Tel Aviv University.

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
