# OpenReview forum: "Optimal Rates for Random Order Online Optimization"
_NeurIPS.cc/2021/Conference — NeurIPS 2021 Oral_

### Official Review · Reviewer_ssvj · 2021-07-11

**Rating:** 6
**Confidence:** 3

**Summary:**

The paper studies the problem of random order online convex optimization introduced previously by Garber et. al in 2020. Random order online convex optimization is a variant of standard online convex optimization where the loss functions may be chosen by an adversary (and do not even need to be individually convex), but the order in which they arrive is uniformly random.  This modification allows one to obtain efficient algorithms with stronger guarantees than for true worst-case analysis.

This paper improves upon the guarantees obtained by Garber et. al, getting nearly optimal rates in the general setting that essentially match offline/stochastic rates.  They remove a factor of $d$ and a factor of $1/\lambda^2$ (where $\lambda$ is the strong convexity parameter) from the regret guarantee.  However, their optimal algorithm requires remembering all previous observations (so $O(dT)$ memory).  Thus, the authors also give an efficient algorithm that requires only $O(d)$ memory but sacrifices a $1/\lambda$ factor in the regret guarantee (which is still better than the result in Garber et. al).   Roughly, the authors draw a parallel to stochastic online convex optimization.  The difference in the random order model is that the observations are not  independent but are instead sampled from a set \textit{without replacement}.  However, the authors are able to modify  techniques from the stochastic case to this without replacement setting.

**Main Review:**

The paper is generally well-written and easy to understand.  The proofs seem correct although I did not check all of them carefully.  I think the overall result is ok.   However,  the paper does fall a bit short in that it does not quite close the random order OCO problem.  In particular, the main question seems to be whether there is a linear memory and runtime algorithm that matches the offline rate of $\log T/\lambda$.  Also, while there are some  nontrivial ideas, I found them a bit unsatisfying and not particularly novel or insightful.  The first algorithm just remembers all previous gradients in order to be able to generate an unbiased estimator of the gradient at each step.


Questions for the Authors:
Both of the algorithms are modifications to online gradient descent.  Do the authors believe that vanilla online gradient descent does not achieve $\log T/\lambda$ or are the modifications more for ease of analysis?

Random order OCO feels quite similar to the stochastic model.  Do the authors have any concrete examples where the random order model would be appropriate but the stochastic model does not suffice?

After Author Response: Thanks for the clarifications!  My opinion remains that this paper is a solid, but not quite spectacular contribution.






**Time Spent Reviewing:**

2

---

> ### Author Response · Authors · 2021-08-10
> **Authors Response**
>
>
> Thank you for your review and taking interest in our paper. We hope that the below will help in clarifying your concerns.
>
> > "the paper does fall a bit short in that it does not quite close the random order OCO problem"
>
> While the question of a low memory optimal regret algorithm indeed remains open, we argue that we *have* answered the question of the minimax rate of the random order online optimization problem. By comparison, flavors of Follow-the-Regularized-Leader or Follow-the-Perturbed-Leader that store loss or gradient history for the per-round minimization objective are not considered lesser state-of-the-art in online convex optimization. In addition, our paper presents significant improvements over prior work on the problem, and builds on entirely different techniques.
>
> > "while there are some nontrivial ideas, I found them a bit unsatisfying and not particularly novel or insightful. The first algorithm just remembers all previous gradients in order to be able to generate an unbiased estimator of the gradient at each step"
>
> The unbiased gradient estimates is far from being the main or important part in the analysis.  Rather, we point out to other two principal ingredients in the analysis, of both algorithms: (1) relating the online performance of SGD (or of any algorithm for that matter!) to its offline performance (i.e., convergence rate) using a novel notion of without-replacement algorithmic stability; and (2) although SGD is uniformly unstable in the setting considered, it can be shown to be *on-average* stable in the sense of bullet (1), subject to proper step size choice.
>
>
> > "Do the authors believe that vanilla online gradient descent does not achieve $\log T / \lambda$ or are the modifications more for ease of analysis?"
>
> Indeed, it is a very interesting question for future work whether vanilla SGD attains the optimal regret (which we now know to be $O((G^2/\lambda)\log T)$!).  At this point, and despite our best efforts, it is unclear whether our inability to answer that question is a shortcoming of our analysis technique (for example, it may be that vanilla SGD is unstable but nonetheless attains the optimal bound), or a result of actual sub-optimality of vanilla SGD for this problem.
>
> > "Do the authors have any concrete examples where the random order model would be appropriate but the stochastic model does not suffice?"
>
> One example that comes to mind, along the lines of the motivation Garber et al (2020) give for the problem, is optimizing media assets (e.g., web page design, ad creatives, etc.) on a landing page for a prescription based internet service opening up in a new country (e.g., Netflix). Since the service is prescription based, every user arrives at the landing page a limited number of times in a limited time span, then either signs up to the service, or decides not to. In any case, the vast majority of users will not return again to the landing page. The above scenario fits the random order model (with user engagement as the loss function) much more faithfully than to the i.i.d.~or adversarial assumptions.
> Having said that, we consider the model and techniques presented in the paper as having the potential of contributing down the line to a grander problem in a way that is only evident in hindsight, as is commonly the case in fundamental research.

---

### Official Review · Reviewer_4xgq · 2021-07-13

**Rating:** 8
**Confidence:** 3

**Summary:**

In this paper the authors tackle the problem of online optimization with functions (picked by an adversary) arriving uniformly at random. Moreover, instead of assuming each function is (strongly) convex individually, they assume only that the sum of these functions is (strongly) convex, as recently proposed in previous work. Their main results are two algorithms for the random order model with strongly convex cumulative loss function, one of them attaining regret matching the assymptotics of the traditional adversarial case and the second only paying an extra $1/\lambda$ factor (both up to lower-order terms) but with a smaller memory footprint. As a corollary they obtain two algorithms for the plain convex case with regret bounds on the order of $\tilde{O}(\sqrt{T})$ and  $\tilde{O}(T^{2/3})$, respectively. To obtain these results they extend the connection between generalization and stability to the "out-of-sample"/sampling without replacement case, which may be interesting by itself.

**Limitations And Societal Impact:**

Yes.

**Main Review:**

# Strengths
- Clear (and significant) contribution to the field, with simple algorithms, an interesting analysis, and with bounds matching the ones of the adversarial case;
- Although dense, the paper is clear and well-written;
- The technique used is by itself interesting and may be of great interest for researchers in related fields.

# Weaknesses
- The lower-bound that justifies optimality does not assume (at least explicitly) smoothness of the functions. Maybe discussing a bit how smoothness does not affect (or how it does affect) the lower-bound would be helpful;
- (Minor problem) some comments before some claims/result statements would ease the task of parsing the results;

# Detailed comments

The results of the paper are very relevant with clear improvements on previous results, the techniques are elegant, and the algorithms are surprisingly simple (the last one in particular is so simple it looks wasteful, although the analysis and discussion make it clear why doing nothing near the end of the game may be useful). That is why I consider this paper to be very strong and a clear accept.

The point I was more confused with was the claim of optimality of the regret bounds for the strongly convex case. In particular, the lower-bound from Hazan and Kale (2011) does not explicitly assume smoothness of the functions. I didn't have the time to check the details on their lower-bound, but I believe this shouldn't be a problem. Yet, I think a discussion regarding this would be interesting.

Finally, this paper is very dense, so using some of the extra-space in the camera-ready version to ease readability would be very helpful.  I have only a couple of suggestions, but you should use your better judgement to improve the readability in the way you think would be best.

- Do the functions need to be 2-times differentiable? Assumption 3 makes it seem like it, but I've not seen the hessian being used explicitly anywhere (even in the appendix). I may have missed some use of it in some proof in the appendix, but I think you should make it clear whether the functions are doubly differentiable or not;
- Typo on line 170 (relvant);
- The notation for filtrations is a bit messy. Filtration is the ordered set of $\sigma$-algebras, not a single $\sigma$-algebra as you make it sound at some points. Defining a $\sigma$-algebra as a sequence of random variables is weird, so maybe explicitly show that it is the $\sigma$-algebra generated by these random variables would be better (that is, $\mathcal{F}_t = \sigma(z_1, \dotsc, z_t)$ instead of $\mathcal{F}_t = (z_1, \dotsc, z_t)$ ). This would be explicitly useful when you have to differentiate between the set of random variables and the $\sigma$-algebra;
- A brief discussion before or after Lemma 3 of what is the role of $i$. At some point we realize it is the point that is certainly going to change in the dataset, but since it doesn't appear very explicitly in the results of the lemma, a discussion of what is the role of this $i$ would be helpful;
- Reiterate either in sec 1.1 or right before the statement of the logarithmic regret result why your regret bound is asymptotically optimal. You mention this on  sec 1.3, but this is an important point worth repeating and going into details about (I was briefly confused whether the lower-bounded cited worked for this setting, but the authors do mention that it holds even for i.i.d. funcionts.)

**Time Spent Reviewing:**

4

---

> ### Author Response · Authors · 2021-08-10
> **Authors Response**
>
>
> Thank you for your review and the points you have raised for improvement. We were happy to hear you enjoyed the paper and appreciate our contributions. Below, we address your questions and suggestions individually.
>
> > "The point I was more confused with was the claim of optimality of the regret bounds for the strongly convex case. In particular, the lower-bound from Hazan and Kale (2011) does not explicitly assume smoothness of the functions"
>
> The lower bound construction in Hazan and Kale (2011) uses quadratic losses, $f(x) = (x - X)^2$ with $X \in \{0, 1\}$, which are indeed smooth (with parameter $2$, same as strong convexity parameter). We will be sure to make this point more explicit in the final version of our paper.
>
> ### Detailed comments
> - *"Do the functions need to be 2-times differentiable?";* This is a good point---no, they do not. We will change the strong convexity definition used in Assumption 3 to one that does not require the Hessian. Thanks!
> - *"The notation for filtrations is a bit messy";* Agreed, will be fixed along the lines you have suggested.
>
> - *"A brief discussion before or after Lemma 3 of what is the role of $i$.";* Good point, we will do so.
>
> - *"Reiterate either in sec 1.1 or right before the statement of the logarithmic regret result why your regret bound is asymptotically optimal.";* We will be sure to make this point clearer. Indeed Hazan and Kale (2011) prove the $\Omega((G^2/\lambda)\log T)$ lower bound for quadratic (hence, smooth) losses which are i.i.d.~distributed (and therefore also applies for the smooth adversarial setting, of course).

---

### Official Review · Reviewer_JxoJ · 2021-07-15

**Rating:** 8
**Confidence:** 3

**Summary:**

Problem description: The paper studies the problem of online convex optimization in the random order model. This problem was recently proposed by Garber et al. (ICML 2020). It is a generalization of online convex optimization, where at each step the learner incurs a loss function $f_t$ and the goal is to minimize the cumulative loss $\sum_{t=1}^{T} f_t$. The authors inspired by Garber et al. (2020), considered the scenario in which each $f_t$ is smooth and Lipschitz but not necessarily (strong) convex. However, they imposed this assumption that $\sum_{t=1}^{T} f_t$ is (strongly) convex. Garber et al. (2020) assumed that at each step $t$, a new loss function $f_t$ is sampled uniformly without replacement from $\{f_t\}_t=1^T$, and they call this setting random order online optimization. The same setting is peruse by the authors of this paper and they improved the results in Garber et al. (2020), under the same assumptions.

Assumptions: 1. $f_t$ is Lipschitz
2. $f_t$ is smooth
3. $\sum_{t=1}^{T} f_t$ is $\lambda$-strongly convex.

Approach: Their main idea is to use the stability properties of stochastic gradient descent, see Shalev-Shwartz et al. (JMLR 2010). The authors defined a new notion of stability (page 5, line 176). This new definition of stability is suitable for sampling without replacement, which is beneficial for random order online optimization. Note that relevant definitions of stability Shalev-Shwartz et al. (2010), works only for i.i.d sampling.

By this conviction, they proposed Algorithm 1 (page 7) which is inspired by Vitter (1985). They proved the stability property of Algorithm 1 in Lemma 6 (page 7), which implies their main result in Theorem 1 (page 8). Since Algorithm 1 requires an intense memory usage, they proposed Algorithm 2 (page 9), which is more economical in that regard. Theorem 2 (page 9) uses Algorithm 2 but with a worse dependency w.r.t $\lambda$ (strong convexity parameter), compare to Theorem 1.

Contributions: With a different and a novel approach, they studied the problem proposed by Garber et al. (2020) and they could improve the results which I summarize as follows:

1. The upper bounds in Garber et al. (2020) have a linear dependency with respect to the dimension of $f_t$. They could remove this dependency and the rates are dimension free.

2. In Theorem 1 (page 8) they improved the results in Garber et al. (2020) up to a factor $1/\lambda^2$

3. In Theorem 1 (page 8) they improved the results in Garber et al. (2020) up to a factor $1/\lambda$

4. As a corollary (page 9) of Theorems 1 and 2, they could obtain similar results for the case when $\sum_{t=1}^{T} f_t$ is convex but not necessarily strongly convex.

**Limitations And Societal Impact:**

The authors briefly discuss the limitations of their work. I also pointed out some, along with suggestions to improve the paper in the Main Review section.

Since the paper is quite technical, I except no direct social impact at this point.

**Main Review:**

Originality: They proposed a new notion of stability which yields to a novel approach for solving random order online convex optimization.

Quality: 1. The theoretical claims are well supported.

2. Authors mentioned Garber et al. (2020) assumed that each loss function has a Lipschitz Hessian, which I do not see this in that paper. I encourage the authors to discuss this.

3. The quality of the paper is fairly high. Their novel approach and improvements were impressive to me.

Clarity: The article is well written. While the notations were complicated, the authors managed to state their results with clarity and least amount of errors. The technical parts are self-contained and easy to follow. The authors provide an adequate literature review.

I have some questions/ comments/ suggestions, which I encourage the authors to discuss or modify:

Questions:

1. line 499: In Lemma 9, why you did not consider conditional probability and expectation for items (i) and (ii)?
2. In Lemma 9, why item (i) holds for $\delta$? Since in Lemma 3, it is valid for the conditional probablity.
3. line 507: why $\delta' < \delta$?
4. line 510: by the definition of $\tilde{\mu}$, how can you show $\alpha = \tilde{\mu}/(1-\delta')$?
5. line 510: under this line and in the equation, according to question 4 how do you manage to obtain the third equality?

Comments/ suggestions:

1. line 67: $\mathcal{O}$ notation hides additive $\beta$ and $D$, but not $G$.
2. line 357: algorithms ----> Algorithms
3. line 469: under this line and in the equation, the term $\ell_S$ should be $\ell$.
4. line 473: under this line and in the equation, the term $A(S, \zeta)$ should be $A(S)$, according to the definition in lines 461 and 462.
5. line 480: in the last equality it would be nice if authors provided a short explanation about why the exchange of the integral and derivative is possible.
6. line 510: by the definition of $\tilde{\mu}$,  $\alpha \geq \tilde{\mu}/(1-\delta')$. By this, in the equation below the third equality should be inequality.
7. line 588: second inequality should be $\eta_{t_0}$, instead of $\eta_1$.


Significance: With a different method they could significantly improve the results in Garber et al. (2020). I like their idea to use stability properties and their novel definition of stability was quite nice. I couldn't understand some parts I mentioned in clarity section and outlined them as Questions. I'm looking forward to their discussion about these questions.

I would like to thank the authors for their response. The authors addressed my concerns in the rebuttal. After reading other reviews and the authors' feedback, my assessment of the paper remains unchanged.

**Time Spent Reviewing:**

8 hours

---

> ### Author Response · Authors · 2021-08-10
> **Authors Response**
>
>
> Thank you for your review and for the thoughtful comments and suggestions. We were happy to hear you appreciate our contributions. Below, we address your questions and suggestions individually.
>
> > "Authors mentioned Garber et al. (2020) assumed that each loss function has a Lipschitz Hessian, which I do not see this in that paper"
>
> Indeed, Garber et al. (2020) neglect to mention this assumption in the main text (as well as the relevant factor in their bounds). It appears in the supplementary material of their ICML'20 version, Theorem A.3. They make this assumption only in the most general setting they consider (where the losses are not required to take any specific parametric form), which is the setting we study in our work. To be more precise, they assume all batches of losses of certain size have Lipschitz Hessians. In any case, we will try to clarify this confusion more accurately in the final version of the paper.
>
>
> ### Questions
>
> 1. Line 499: Both items (i) and (ii) make statements regarding the conditional probabilities and expectations, this is notated using the subscript $t$, as defined in the beginning of section A.3, right before the statement of the lemma.
> 2. See previous bullet. (We preferred keeping some notation out of the main text, hence the discrepancy of notation between the two lemmas.)
> 3. Line 507: By assumption (i) $\Pr(z_t \neq z_t') \leq \delta$, and $\delta' := \Pr(z_t \neq z_t')$ just denotes this quantity (see line line 505), hence $\delta' \leq \delta$.
> 4. Line 510: There is a small error there - thank you for pointing this out. The correct derivation should read $\alpha \geq \tilde \mu / (1 - \delta')$ (justified by $\delta' \leq \delta$), and then in the second to last line of the multiline equation, where there is an equality, it should be a $\leq$; justified since $-\alpha\leq -\tilde \mu / (1 - \delta')$.
> 5. See previous bullet.
>
> ###  Comments/Suggestions
> 1. It hides factors *additive* in $G$, and does not hide factors *multiplicative* (relative to the $T$ term) in $G$. We will amend so that this is clearer.
> 2. Thank you, will fix.
> 3. Right, nice catch.
> 4. Right, we hoped that the intention is clear and sacrificed a bit of formality. Will fix.
> 5. We will add an explanation in the final version as you suggest.
> 6. Indeed you are right (see answer to question 4).
> 7. Assuming you meant line 568 - correct, will fix.

---

> > ### Comment · Reviewer_JxoJ · 2021-08-25
> > **Aknowledging**
> >
> > I would like to thank the authors for their response. The authors addressed my concerns in the rebuttal. After reading other reviews and the authors' feedback, my assessment of the paper remains unchanged.

---

### Official Review · Reviewer_zUYG · 2021-07-16

**Rating:** 7
**Confidence:** 2

**Summary:**

The authors study a variant of OCO introduced by Garber, Korcia, and Levy in which the costs are adversarial but are revealed to the online learner in a uniformly random order. The authors propose a new gradient-based algorithm for this model and claim that the regret incurred by their algorithm has a strict improvment on the problem parameters compared to that of Garber et al. Specifically, the regret of the new algorithm does not depend on the ambient dimension $d$ at all and has improved dependence on the strong convexity parameter $\lambda$.

**Ethical Concerns:**

No problem here.

**Limitations And Societal Impact:**

No problem here.

**Main Review:**

This paper studies an interesting problem and is well-written.  The main two contributions are 1) the paper obtains the optimal minimax regret rate in random order online optimization, significantly improving on the work of Garber et al, and 2), the paper introduces a new stability analysis of gradient descent, which will almost certainly be of broader interest to the learning theory community. The connection between regret minimization and generalization theory is very cool and I am sure will be the focus of much future work.


Verdict: The results and proof technique presented in this paper are interesting and novel. After reading the other reviewer's comments and the author responses, it is clear that I misunderstood the results of this paper in my initial review. I am increasing my score accordingly.

**Time Spent Reviewing:**

2

---

> ### Author Response · Authors · 2021-08-10
> **Authors Response**
>
>
> Thank you for the review. We hope that the below will help in clarifying your concerns.
>
> > "the authors of this paper make an additional structural assumption about the costs: each function is identical, except that it is evaluated at a different datapoint $z_i$"
>
> We **do not** make any assumption on the parametric form of the loss functions, and accordingly **do not** make use of such an assumption in any of our proofs. Having a single loss function $f$ evaluated at different datapoints $z_i$ is simply a matter of notation: we could have used a notation of the form $f_i(w)$ rather than $f(w,z_i)$ to refer to different loss functions, but chose the latter alternative since we frequently evaluate expectations with respect to the datapoints/indices. For the example you mention of losses $x^2, x^4, \log x,$ etc., you may use the datapoint $z_i$ simply as an index into a set of $T$ losses, e.g., $f(x; 1) = x^2, f(x; 2) = x^4, f(x; 3) = \log x$, and so forth. We stress that this convention is common and fairly standard; e.g., see the classic paper "Stochastic Convex Optimization" by Shalev-Shwartz et al (COLT 2009), where the exact same notation is used.
> If there are any concerns with a particular argument we will be happy to clarify further during the discussion.
>
> > "I ultimately feel that the authors present their work in a misleading way, failing to mention that improvement of their results relative to those of Garber et al potentially arise from much more restrictive assumptions about the cost sequence."
>
> We have made every effort to make the relation of our results to those of Garber et al. (2020) as clear and fair as possible. We reiterate our response above: we **have not** made any assumptions beyond those of Garber et al. (2020); in fact, we have *removed* one assumption - that of Lipschitz Hessians (see the supplementary of their ICML'20 version, Theorem A.3.).

---

> > ### Comment · Reviewer_zUYG · 2021-08-20
> > **My mistake!**
> >
> > It is clear that I misunderstood the setting considered in this paper. I sincerely apologize to the authors and the other reviewers for this oversight! I have changed my score from reject to accept.

---

### Decision · Program_Chairs · 2021-09-27

**Decision:**

Accept (Oral)

**Comment:**

The paper eventually received uniformly positive evaluation, with two of the reviewers rating it to be within top 50% of accepted papers.

There is clear and significant contribution to the random-order Online Convex Optimization, with the results improving upon previously known by Garber (2020), and getting nearly optimal rates in the general setting.